# A Multirobot System in an Assisted Home Environment to Support the Elderly in Their Daily Lives

**DOI:** 10.3390/s22207983

**Published:** 2022-10-19

**Authors:** Ramón Barber, Francisco J. Ortiz, Santiago Garrido, Francisco M. Calatrava-Nicolás, Alicia Mora, Adrián Prados, José Alfonso Vera-Repullo, Joaquín Roca-González, Inmaculada Méndez, Óscar Martínez Mozos

**Affiliations:** 1Robotics Lab, Universidad Carlos III de Madrid, Avda. de la Universidad, 30, 28911 Leganés, Spain; 2Department of Automation, Electrical Engineering and Electronics Technology, Universidad Politécnica de Cartagena, St. Dr. Fleming, s/n, 30203 Cartagena, Spain; 3AI for Life, Centre for Applied Autonomous Sensor Systems (AASS), Örebro University, 70281 Örebro, Sweden; 4Department of Evolutionary and Educational Psychology, Faculty of Psychology, University of Murcia, 30100 Murcia, Spain

**Keywords:** assistive robotics, social robots, aging, Ambient Assisted Living (AAL), Node-RED, IoT, interoperability, heterogeneous systems, ROS, smart home, robotic manipulation, well-being

## Abstract

The increasing isolation of the elderly both in their own homes and in care homes has made the problem of caring for elderly people who live alone an urgent priority. This article presents a proposed design for a heterogeneous multirobot system consisting of (i) a small mobile robot to monitor the well-being of elderly people who live alone and suggest activities to keep them positive and active and (ii) a domestic mobile manipulating robot that helps to perform household tasks. The entire system is integrated in an automated home environment (AAL), which also includes a set of low-cost automation sensors, a medical monitoring bracelet and an Android application to propose emotional coaching activities to the person who lives alone. The heterogeneous system uses ROS, IoT technologies, such as Node-RED, and the Home Assistant Platform. Both platforms with the home automation system have been tested over a long period of time and integrated in a real test environment, with good results. The semantic segmentation of the navigation and planning environment in the mobile manipulator for navigation and movement in the manipulation area facilitated the tasks of the later planners. Results about the interactions of users with the applications are presented and the use of artificial intelligence to predict mood is discussed. The experiments support the conclusion that the assistance robot correctly proposes activities, such as calling a relative, exercising, etc., during the day, according to the user’s detected emotional state, making this is an innovative proposal aimed at empowering the elderly so that they can be autonomous in their homes and have a good quality of life.

## 1. Introduction

The increasing isolation of the elderly both in their own homes and in care homes has made the problem of caring for elderly people who live alone an urgent priority. Several experts have spoken in favour of “the availability of comprehensive and integrated care at home” [1]. During the COVID-19 pandemic, the care-response capacity in nursing homes was overwhelmed, causing a dramatic situation. For this reason, the protection of the elderly in Spain now presents a different scenario that requires person-centred care, for which adapted housing can be used [2].

It will be difficult to fully meet the demand for home care in the near future due to the shortage of available health personnel that is a result of increased life expectancy and low birth rates. Spain continues to head the list in terms of longevity in Europe, with an average longevity of 82.4 years. Unfortunately, aging often goes hand in hand with deteriorating health, and, in many cases, this requires additional attention. The number of those over 65 who live alone has increased by 30% in the last ten years, so that it can be concluded that people in Spain, and in most first-world countries in general, are living longer and have more health problems, in addition to the growing tendency to live alone.

In addition to the psychological problems derived from the growing tendency to loneliness, the progressive deterioration of physical capacities with age must be considered. Everyday tasks, such as getting up to move the shutter or answer the phone, fetch a glass of water or hot milk from the kitchen, can be difficult challenges to overcome. Current technology should be able to provide solutions to help people living alone with these needs.

In the last five years, technology has invaded almost all areas of our lives; it helps us at work, in our leisure time and in answering medical questions. However, there is another important aspect that has not been given the attention it deserves and which is now gaining prominence: the care of elderly people in their own homes. Although solutions have been found for specific problems, at present, there are no generic, coordinated, adaptable and affordable solutions, as stated in [3].

In this regard, we are faced with a scenario in which the environment must relate to the person, be adaptable to the user’s capabilities and help him/her to be autonomous with a better quality of life. For this, we propose three coordinated actors to interact with the person:The home itself, i.e., the immediate environment: turning lights on and off and moving the blinds without having to get up, having technical alarms in the home (gas leakage, water, fire detection) to detect dangerous situations, such as a stove that has been left on, but also collecting information through non-invasive and cost-effective commercial sensors, which, thanks to artificial intelligence, can detect situations in which help is necessary and suggest activities to improve mood. A wearable on the wrist collects personal data for the training of a machine-learning algorithm to predict the user’s mood.A social mobile robot that interacts actively with the user. This robot is equipped with an openable tray to carry objects weighing less than 1 kg.A two-armed domestic-assistance robot to manipulate objects for the service robot, performing basic actions, such as picking up objects from the kitchen, manipulating them and leaving them on the service robot’s tray.

These elements constitute a heterogeneous system called HIMTAE (Heterogeneous Intelligent Multirobot Team for the Assistance of Elderly People) for the assistance of elderly people who seek comprehensive care. The main contribution of this work is the proposal of this heterogeneous system that includes elements from different disciplines, such as social and assistive robotics, psychology, home automation, Ambient Assisted Living (AAL) and artificial intelligence (AI), which aims at analyzing a subject’s behaviour and state of mind and suggesting emotional coaching activities through the user’s own smart devices. The proposed Multirobot and Ambient Assisted Living system is modular and scalable according to the user’s needs, thanks to its open architecture. Different system deployments are possible, from the simplest model (home automation and smart speakers to communicate and interact with the user’s environment, possibly including the social robot) to the most complex model, which includes two robots, i.e., a social assistant and a manipulator. The small mobile social robot monitors the user’s well-being, giving suggestions to improve their mood, while the domestic manipulating robot helps to perform household tasks. These types of integrated platforms with home automation solutions aligned with Ambient Assisted Living (AAL) belong to the field of social and assistive robotics to improve people’s everyday lives.

Both robots have different requirements, morphological characteristics and functionalities, depending on the tasks to be performed. The companion social robot must be small, user-friendly, capable of interaction and possess the software and algorithms required to process the types of information necessary. The domestic-assistance robot must be larger to have access to the work areas, be able to carry loads and be equipped with handling capabilities. Its manipulation capabilities must include planning algorithms and environmental modelling in both 2D and 3D.

Previous works, such as [4], have proposed solutions based on multirobot systems for people with reduced mobility, but these have consisted of an accompanying robot and an autonomous wheelchair without a manipulation robot for physical assistance. In [5], a multirobot system was proposed which consisted of a multirobot system that included serial manipulator arms, a wheeled base and an approachable human-like face but which did not incorporate Ambient Assisted Living (AAL). Other proposed solutions include multirobot systems [6,7], but in these cases the platforms are social robots without manipulation or movement capabilities. In [8], the authors present a multirobot system based on cloud computing to share information but which is not capable of performing tasks that require manipulation. The present proposed system consists of a multiple robot system with AAL and manipulation capabilities designed to provide complete assistance (mental and physical). Both platforms were considered to need movement capabilities to follow and monitor the companion, while a domestic physical-assistance robot needs to be able to position itself in the work area and any part of the house.

In the following two subsections, other state-of-the-art solutions are presented; in Section 1.3, the HIMTAE system overview and its main contributions are described.

### 1.1. Related Works on Assistive Robots and Ambient Assisted Living

Recent developments in companion robotics suggest that social robots could be used successfully in elderly populations to improve mood, decrease negative emotions and reduce loneliness [9,10,11]. Studies on elderly populations have used pet-like robots, such as Paro, Aibo and NeCoRo. Samsung’s BotCare [12], Rassel robot [13] and Mini robot [11], although not mobile, can focus on interactions and mood recognition. The idea of completing a system with a multirobot care team is proposed in [14], which defines the theoretical aspects of integration. In [15], the Romeo2 project introduced the concept of an assistant robot for everyday life in the form of a humanoid robot, although it focused more on interaction with the user. Ref. [16] proposed a manipulator to assist and entertain elderly people living in apartments, but it has limitations regarding manipulation and operations in human environments. In addition to these systems, new designs can be found in the recent literature, such as [9], which describe the current state of the art in assistive robots to help older adults through the COVID-19 pandemic. Further examples are presented in [17] of how assistive robots could improve the elderly’s independence as regards their reduced mobility, including rehabilitation needs.

However, as the range of psychological problems that previous work has addressed is limited, recent analyses have urged greater efforts in robotics to treat common mental disorders, such as depression and other mental health problems associated with high individual and societal costs.

Robotic systems are now able to interact in more complex ways with human users, offering new opportunities to provide individualized healthcare and mental care. Much attention and research are being devoted to assistive systems that aim at promoting independent living at home for as long as possible. The transition from home automation to the more ambitious Ambient Assisted Living (AAL) paradigm has given the opportunity to integrate robots and ambient home help systems. An example of the symbiosis between AALs and robotics is given in the H2020 ENRICHME project, in which environmental sensors provide information on activities to an assistive robot. Refs. [18,19,20] detail several studies of smart environments and robotic assistive technologies that have the potential to support the independent living of older adults at home by implementing age-friendly care services. Recent successes of the AAL program include ReMember-Me (https://www.rememberme-aal.eu/es/inicio/ (accessed on 10 September 2022)), an intelligent assistant to detect and prevent cognitive decline and promote social inclusion. This is a technological solution composed of a social robot in a tablet plus smart sensors. Through the system, users will be able to perform various exercises and assessment games tailored to their performance level and preferences, receive personalized health recommendations and share their knowledge with reputable members of the community. Their progress will be recorded and displayed to family members and authorized healthcare professionals to aid early detection and intervention. In Spain, some research groups are promoting the use of social robots for physical and cognitive stimulation in elderly care facilities with success, as in Zalama [10], which puts forward the innovative idea that mood levels can be estimated from automatic observation of Activities of Daily Living (ADL) by integrating AAL sensor data with robot sensors. Salich [11] and other groups working on European projects, such as Pharaon (www.pharaon.eu (accessed on 10 September 2022)), are more concerned with healthcare but deal with active and healthy aging aided by technology [21].

The HIMTAE system proposes a two-armed manipulating robot capable of working in a home environment in coordination with a small mobile assistant robot. In this field, the ability to manage a team of heterogeneous robots is desirable. In [22], a multirobot task-planning-and-execution architecture is presented. The system is designed to coordinate a team of mobile robots with different physical characteristics and functions. The system also allows interaction with multiple users. People with reduced mobility can also benefit from approaches based on multiple assistance robots, as in our proposed system, where a small robot can collaborate with another manipulator robot to transport small objects to a person with reduced mobility or help with the previously mentioned household jobs that include the use of robotic wheelchairs [4,5,17]. Research is emerging to coordinate these robots in smart places, as in [14], with an approach aimed at enhancing Ambient Assisted Living services. This work aims to integrate a Robot-Assisted Ambient Intelligence (RAmI) architecture with mobile robot teams. The RAmI architecture contains a semantic, a scheduling and an execution layer.

Regarding the advanced architecture of this type of system, it is important to achieve persistent autonomy, including planning, reasoning and acting phases. In this regard, automated planning for high-level control of robotic architectures is becoming widespread, thanks mainly to the capability to define tasks to be performed declaratively [23]. The authors of the latter study propose a classical planning system and its extension to the action-based planning paradigm. This allows the robot’s actions and the possible states of the system to be defined using the Planning Domain Definition Language (PDDL) [24], a predicate-logic-based language. PDDL is also used by other systems for AI planning and reasoning in socially assistive robots, as in [25], which proposes a modular and common framework and adopts two standards: the ROS platform and the PDDL language.

Regarding task planning, there are other methods proposed by different authors. One method is a marks-based approach. For instance, in [26], tasks are regarded as trees, in which subtasks can be contracted to different agents. Another method is a mixed-integer linear program, as proposed, for example, in [27]. In [28], a method for simultaneous task allocation and planning (STAP) is presented to combine the planning of tasks with their allocation to agents. Another algorithm used for path planning is fast marching. In [27], the method is applied twice (FM^2^). The solution can deal with both individuals and groups using O-space values as indicators for the robot’s social behaviour. This solution allows the robot to navigate an environment populated by humans who may be obstacles that need special treatment and who at other times may be the objectives. A generic framework for social path planning can be studied in [29], and an implementation of a dynamic path-planning method for social robots in the home environment can be found in [30].

Grasping assistance may be needed by elderly people who have reduced mobility and motor deficiencies and need solutions to various problems. In [31,32], a proposal is made to facilitate human–robot interaction in gripping tasks. This involves scene segmentation, prehension and recognition of 3D objects. They authors present a method based on decision trees for object prehension.

The last issue also requires the detection of objects and place recognition. There are different ways to approach this problem. In [33], a lightweight convolutional neural network (CNN) approach is proposed for visual place recognition.

The development of pre-grasping and grasping techniques that include object recognition, path planning and learning methods are important contributions to grasping tasks.

### 1.2. Related Works on the Assessment of Emotional States, Mental Health Monitoring and Coaching

The field of mood recognition has gained in importance in the world of artificial intelligence. Successful studies have been conducted with wearables [34,35], such as IoT sensors for behavioural inference [36]. However, putting the system to practical use is still the challenge we are facing with the proposed system.

Different sensor systems have been proposed to collect physiological and activity data and analyze them to automatically assess people’s moods. Smart monitoring systems use smartphones and self-reported depression reports to assess people’s moods, as in [37]. However, the latter system does not provide counselling. In the H2020 Help4mood project, data on daily activities are converted into graphical, textual and conceptual summaries that can be communicated to clinicians providing external assessment.

Automatic mood monitoring and assessment are important components of support therapies and daily life coaching. Ecological momentary assessment (EMA) can ease patients’ mental burdens, since the traditional assessment tests are too long to be used repeatedly. In addition, the latest smartphones and wearable devices make the EMA approach much more feasible as a solution for monitoring mental illness and offer economically friendly solutions.

The above-described method of gathering information on mental well-being has been used in several works. In [38], a system is proposed to monitor potential depressive patterns in elderly people living alone. Emotional states are assessed from diverse sources, such as surveys, smart watches and EMA questionnaires. EMA methods for obtaining emotional information can expand this research area to practical applications. For example, in [39], a system which gathers information on a person’s activities can detect long-term stress patterns. Stress detection during daily real-world tasks through advances in intelligent systems is described in [21]. Techniques such as machine learning were also applied in [19] to monitor elderly people’s moods via intelligent sensors on wristbands.

An issue to be addressed is the usability of these modern assistive technologies by the elderly. In [40], the authors present a user-centred design for a web-based multimodal user interface tailored for elderly users of near-future multirobot services.

### 1.3. HIMTAE Overview and Main Contributions

The integration of a heterogeneous multirobot system in a domestic environment is an important contribution of the HIMTAE system for domestic tasks. It considers both mental assistance in an AAL environment and robot task path planning and grasping problems to assist humans in their tasks. It also employs robot–user interaction with mental health monitoring and coaching to interact with humans. This HIMTAE system extends IoT sensors with distributed artificial intelligence to strengthen the symbiosis between AAL and robots. Our system proposes a change in training to address different mood episodes and will advance the “interaction capability” of robotics in care scenarios, developing novel techniques to cue users and engage them in therapeutic activities.

The development of 3D planning task algorithms is needed for mobile manipulators that include safe 2D displacements and 3D plans for manipulation tasks, considering the optimal assignment of tasks to be performed by the robots. It will take account of the heterogeneous system’s resolution of navigation problems, the occurrence of unforeseen events, people, etc.

The HIMTAE system extends mental assessment and mood monitoring with proactive coaching capabilities, provided by a companion social robot, which suggests activities to older people to improve their mood and mental well-being. Following the WHO’s [41] recommendations on ethics and governance in the field of health, artificial intelligence, as used in this work, holds great promise for public health practice, as it has great benefits, such as protecting human autonomy, promoting the well-being and safety of individuals and the public, ensuring transparency, clarity and intelligibility, and guaranteeing inclusiveness, equity and sustainability.

The proposed HIMTAE heterogeneous robotic and AAL system comprises several components, summarized below and described in detail in the following sections. It should be remembered that different deployments of the system are possible, as mentioned previously, from the simplest to the most complex model, including both robots. Using low-cost commercial components (such as low-cost commercial sensors), open-source software (ROS, Node-RED v.3.0.2, Home Assistant 2022.10.3) also favours the incorporation of new hardware and software for future developments.

Assistive mobile robot: Designed to navigate autonomously around the house to attend to the users’ needs or suggest an activity according to their mood.Home automation sensor ecosystem: Includes a set of low-cost sensors, both commercial and self-designed, to monitor the user’s lifestyle.Empatica E4 medical device: Monitors certain physiological variables, such as BVP (blood volume pulse), EDA (electrodermal activity), skin temperature, HR (heart rate), IBI (interbeat interval) and acceleration, for the training of a machine-learning algorithm to predict the user’s mood.Application for psychological data acquisition: Carries out a psychological study and thus relates the information obtained from the Empatica E4 to the user’s state of mind, such that basic questions about activity/well-being can be answered. This is another input for the machine-learning algorithm.Domestic-assistance manipulative robot: Carries out domestic tasks which require both the ability to move around a room and handling skills. This creates a need for a larger mobile platform than the companion robot, similar in size to a person, since it must adapt to the household dimensions.Central home-assistant unit: Collects home automation data on the person’s routines at home and activates the necessary devices. This unit, currently a Raspberry Pi, will be replaced by an embedded PC to house the artificial intelligence algorithms for the coordination and functioning of the various well-being devices. ROS, Home Assistant and IoT were used to make the information transparent among the system’s different elements.

Figure 1 shows a scheme of the main elements.

## 2. Materials and Methods

Several technologies and methods for systems integration that have been used to build the heterogeneous system described above and represented in Figure 1 are described in this section.

Section 2.1 describes how the ecosystem of sensors and actuators distributed throughout the home has been designed as an Ambient Assisted Living (AAL) system to monitor and care for the elderly, as well as the acquisition of physiological signal data and the estimation of mood states using machine-learning algorithms, including the psychological tools used.

Section 2.2 describes the two coordinated robots designed for assistance in household tasks, the domestic manipulator and a small low-cost social robot that attends to the user’s requirements and proactively suggests emotional coaching activities according to the mood it perceives by means of the estimations made by the AI and Ambient Assisted Living systems.

Section 2.3 describes the methods used for the modelling, planning and navigation of the robot’s movements and routes, using state-of-the-art and self-development algorithms.

Section 2.4 describes the software architecture proposed to integrate the heterogeneous systems that make up HIMTAE, as well as the communication techniques and protocols respecting the different nodes.

### 2.1. Ambient Assisted Living and Data Acquisition

Although many research efforts have focused on the development of home automation and elderly care systems with service robots in the context of home care, it is currently difficult to find near-market solutions for home automation and remote care for the elderly at home, including affordable assistant robots and Ambient Assisted Living (AAL) capable of analyzing the environment, determining subjects’ emotional states and acting accordingly, and interacting with users in a natural way, adapting to them according to their roles (elderly patients, relatives or caregivers), among other characteristics. The home automation components that guarantee AAL in this system are minimally intrusive.

#### 2.1.1. Sensor Ecosystem

Figure 2 provides a complete proposal for the distribution of basic commercial home automation sensors, while simplified schemes of this deployment are adopted in the test scenarios, which will be described below.
The main activities that can be monitored so that, with the support of the psychology team, emotional coaching strategies can be proposed are given below; further information can be found in [36]:Hours of sleep, physical activity inside the home, the times the subject visits the bathroom and time spent in different rooms. For this, activity bracelets are used, such as the Empatica E4, and movement sensors, such as Xiaomi’s Mi Motion Sensor;Time sitting in front of the television or in bed using simple seat sensors with an electronic interface developed for the proposed system;Alteration of habits or mood due to changes in thermal sensation and light intensity through temperature and light sensors;Cleaning activities, organization, use of the refrigerator, etc., registered through opening detectors, electrical consumption sensors, etc;Stress and activity levels and other biometric parameters to supplement mood estimation by the Empatica E4 medical bracelet [42].

As all the commercial sensors use the Zigbee protocol, to manage them locally with the Home Assistant, the generic CC2531 gateway was used. The option of using the Conbee II gateway was also originally considered but was discarded for stability reasons. A low-cost ESP32 microcontroller [43] was used for the self-designed sensors to transmit their information through the MQTT communication protocol to a Mosquitto broker mounted on the same Raspberry Pi system as the Home Assistant.

#### 2.1.2. Data Acquisition of Physiological Signals and Estimation of Mood State

One of the HIMTAE’s main features is that it analyses physiological and psychological data to estimate the subject’s mood, with the help of AI [44]. Although it is true that good results have been achieved in this field, many of the tests have been performed in laboratory environments [45,46], i.e., the entire process has been under constant surveillance. The proposed system aims to provide a way of capturing data that does not intrude in the user’s daily routine and does not require constant surveillance.

Physiological and activity measurements are taken by means of an E4 medical wristband from Empatica, which monitors certain variables, such as BVP (blood volume pulse), EDA (electrodermal activity), skin temperature, HR (heart rate), IBI (interbeat interval) and acceleration in three axes (*XYZ*).

The different physiological signals are first preprocessed and filtered. Features of all the signals are extracted using a sliding window approach, with a window size of 60 seg. We also set an overlap of 10% between consecutive windows to reduce the boundary effect when signals are filtered. The process of feature extraction from the various sensors included in the Empatica e4 wristband is detailed in [44]. Features in both time and frequency domains are calculated for each component. These will be part of the dataset necessary to train the machine-learning algorithms to predict the user’s mood. In particular, several tables in [44] show 72 features from the 3-axis accelerometer (minimal and maximal values, standard deviations, etc.), 13 features of peripheral skin temperature (means of absolute values, mean value of power spectral density, etc.), 27 features of heart-rate variability (high-frequency spectral power as a percentage, etc.) and 82 features of electrodermal activity (means, standard deviations, etc.).

Taking the experiments described in [44] as a starting point, as were taken into account in [42], an Android application was designed to capture psychologically relevant data (Figure 3). The system uses 2 tools, the first of which consists of EMAs (ecological momentary assessments) based on the mental/mood state model proposed by JA Russell [46], such that the user is asked 5 times a day about their levels of happiness and activity, dividing these parameters into 5 different discrete levels ranging from 0–4. The second tool used is a questionnaire made up of the first 20 questions of the STAI questionnaire, which provides potential insights into symptoms of anxiety and depression. Frequency analysis was carried out to contrast the correlation between the arousal (or activeness) and pleasure (or happiness) dimensions with the anxiety levels registered (Table 1). The results concluded that correlations between affective dimensions and anxiety states were not statistically significant [44].

As shown in the scheme in Figure 3, the method of estimating the person’s state of mind at home is based, first, on a comparative study of the user’s own perceptions, with their states of happiness and activity marked in the mobile application and physiological data captured by the Empatica medical device. When enough user data have been obtained, the pertinent characteristics of the physiological data are obtained for labelling with the psychological data for subsequent classification by a classifier built using the SVM (support vector machine) libSVM library (www.csie.ntu.edu.tw/~cjlin/libsvm/ (accessed on 10 September 2022)). Once the model has been built and sufficiently trained, the final goal is the estimation of the user’s mood without them having to answer the questionnaires [44].

### 2.2. Mobile Assistance and Domestic Manipulator Robotic Platforms

The proposed system involves a multiple robot system that includes AAL and manipulation capabilities to provide fully comprehensive assistance (mental and physical) to the user. Both robots have different requirements, morphological characteristics and functionalities that depend on the tasks to be performed. The companion social robot is user-friendly, with greater user-monitoring and -interaction capabilities. The domestic-assistance robot is larger to enable access to work areas and the support of greater loads and is equipped with handling capabilities.

#### 2.2.1. Mobile Assistance and Coaching Robot

The Turtlebot II low-cost business solution [47] was used as the mobile base for the assistance robotic platform, which, with the proper voice and graphical interface, will be able to propose emotional coaching activities to the person (such as suggesting that they move to another room, stop watching television and exercise, make a video call to a family member (facilitated by the system), etc.). This is a well-known differential-type mobile robot with two support wheels in the shape of a rhomboid. The rest of the robotic platform is mounted on the Kobuki’s IClebo mobile base, which offers the possibility of mounting platforms on top in a modular structure to incorporate the necessary extra hardware, such as the Intel NUC CPU and the LIDAR Hokuyo UST-10LX. To make it user-friendly and with a graphical interface for the user, a 3D design and 3D-printed prototype have been attached on top of the Kobuki base, as shown in Figure 4. The user interface integrates natural language in addition to visual interaction by means of an Echo show 8 smart speaker, with a screen and an Alexa mounted on the top platform.

The robotic platform uses Ubuntu 16.04 as the operating system and ROS as the software robotics framework. The mapping and navigation stack offered by ROS is used to map the home in which the robot is installed and for autonomous navigation once it is mapped. The robot can navigate inside the home, avoiding both static and dynamic objects, thanks to the Hokuyo UST-10LX LIDAR that provides precise information on the environment for mapping and navigation.

#### 2.2.2. Domestic-Assistance Mobile Robot

The domestic-assistance robot is a two-arm manipulative robot that can perform domestic tasks, such as cooking, and has the ability to move around a room as well as manipulation skills. To achieve these functionalities, a larger mobile platform than the accompanying robot was needed, one similar in size to a person that is able to adapt to the dimensions of a home. The platform is designed to interact with people and takes semantic concepts into account in its algorithms and contemplates social path planning [28].

The selected robot is made up of a mobile base, a torso and two arms, reaching a total height of 160 cm. The 50 cm-diameter base is a Robotnik model RB-1 [48] actuated by two motorized wheels and three support wheels that move forward and backward and rotate but which cannot move sideways. The base contains a 2D laser sensor a short distance above ground level, which obtains geometric information from the environment for navigation. The torso is made up of two parts: one on top of the base, in the shape of a cuboid, that contains the arm-control computers. The upper level contains the arms. The first part has three metal trays at the front for the transportation of various objects. The two UR3 arms are from Universal Robots [49], with a range of 50 cm and a load of 3 kg. They have a total of six degrees of freedom each and a range of movement in each joint between −360° and +360°, except for the final effector, which allows more than one revolution. Figure 5 shows a diagram of the different parts that make up the robot.

A gripper for handling tasks was produced with a 3D printer. The apparatus consists of two fingers sub-actuated by a servomotor to open and close the grippers, in addition to two linear impedance sensors in the fingers to detect pressure when grasping an object and prevent the action of the clamp.

To perform its tasks, the device is equipped with mobile navigation capabilities, such as those provided by the mobile assistance platform, which is adapted to the size of the mobile manipulator for correct positioning in work areas. A model of the work area is also needed, as well as planning of the trajectories for arm movements and manipulation.

### 2.3. Methods used for Modelling, Planning and Navigation

The navigation system and the environmental modelling are based on the geometric characteristics of the environment for local navigation but use semantic concepts to facilitate user interaction and social path-planning techniques to move around the environment. The environment is modelled using occupation maps in which free, occupied and unknown zones are identified by the SLAM technique, which positions the robot during exploration of an unknown environment. This process is carried out in the ROS environment through the Gmapping package [50], which takes odometry and laser sensor values as inputs and builds an occupancy map as output. This process is combined with an autonomous scanning algorithm [51].

Due to the robot’s large size, its behaviour must be able to adapt to whatever actions it executes. It has to be able to differentiate between navigation in an unobstructed room and in confined areas, especially doorways, so these must be defined on the occupation map.

The method selected for this was Watershed [52], characterized by its fast execution. Once each of the rooms has been differentiated, a topological map is extracted for navigation between different areas in the house, and semantic labelling is applied based on the geometric zones detected in the environment and the uses to which they are intended by the user. An example is shown in Figure 6, with the initial occupancy map, segmentation of the rooms by colour and the corresponding topological map. Each of the nodes represents the available rooms (R) and doors (D), while the arches (E) indicate the connectivity between zones. This method is applied to both the assistant and the bimanipulator robot.

Once the environment has been modelled, localization and planning techniques are used to navigate with the base. Initially, the locator provided by the ROS navigation stack is used, based on Monte Carlo methods [53]. Given the current location of the robot and the desired destination point, the path to follow is planned in two phases. In the first phase, the topological map is used to check the rooms and doors that must be crossed and passed through. The Dijkstra planner [54] is then applied after identifying the topological nodes of the initial and final points. In the second phase, two different behaviours are alternated depending on whether the robot is to pass through a door or navigate a room.

In the first case, low-level navigation is used, in which the robot first orients itself to-wards the door and then moves in a straight line, using the information from the laser sensor on the base to correct its orientation and avoid colliding with the frame. In the second case, the selection of a path takes into account the user in the environment. To go through a door, low-level navigation is used, in which the robot first orients itself towards the door and then moves in a straight line, using the information from the laser sensor to avoid colliding with the door. When passing through a room, FM^2^ [55] is selected. This algorithm is executed before the robot starts to move, so that an overall plan is obtained. If an unexpected object or a person is detected, it is added to the occupation map and the modified FM^2^ considers the new object or social distance and the task to perform in the case of a person, this being executed again so that the robot can avoid the new element in real time [27].

#### Handling Planning

To plan the trajectory of the arms of the manipulating robot, a simulated model is used to verify that the trajectory will not collide with other elements in the environment (which has been modelled previously). Free and occupied spaces are defined by 3D occupancy maps. Both the robot’s body and the elements in the room are occupied non-traversable spaces with a value of 1, while the rest are traversable spaces with a value of 0. An example of the model used is shown in Figure 7 with a simplified arm structure.

It is then necessary to know the coordinate axes of the parts of the robot in the real system to establish the necessary transformations with respect to our simulated model, for which the 3D Rviz viewer is used. The axes involved in this task are shown in Figure 8. For this, the robot’s basic axis is needed, since all the sensors are referenced with respect to it, along with those of the arms, since the movement commands must be referenced to them. The coordinate axes of both arms are different.

The coordinate system of the arms is rotated with reference to a certain angle with the base, which is 45° in both cases. The arms are displaced around the three axes, *XYZ*, with respect to the base. Both rotation and translation operations will be necessary to establish the relationship between these three reference axes, so that all the points calculated with respect to the robot base by the sensors can later be referenced with respect to the arms, thus avoiding errors in executing a movement. This process is based on a series of transformations through which each of the joint positions are sent to generate the required trajectory in the real robot model [56], for which the following steps are followed:First, the real point in the environment to which to move the end effector of the arm is located, using a RealSense D435RGBD camera to obtain its coordinates in three dimensions.The values of the real environment are converted to the simulated model in Matlab software. This environment (as shown in Figure 8) is based on a simplified model translated with respect to the position of the real robot.The necessary transformations are established between the robot’s base with respect to which points of the real environment are taken and the bases of each of the robot’s arms.The corresponding planner is executed, and the transformations are undone by working with the inverse matrices of the calculated transformations to send the trajectory to the real robot with respect to the bases of each arm.

The generation of the trajectory is then carried out to move the end effector of the arm to the desired point, applying same FM^2^ and modified FM^2^ to take into account objects and users; unlike the base case, this is performed in a 3D environment, based on the concepts described above. Using FM^2^, the trajectory generated for the end effector is obtained. Collisions with objects in the environment are avoided by applying a method based on differential evolution [57], which verifies that the points of the joints and the links between joints do not come into contact with an occupied element or with themselves.

### 2.4. Software Architecture for Systems Integration

ROS middleware was used for the integration of all the elements, robotic platforms, home automation sensors and physiological information. It allows the architecture to be organized in nodes, is multilingual, follows the publisher/subscriber policy and provides flexibility and ease of understanding in terms of design.

Integration of the different elements of the heterogeneous system and ROS was also carried out and validated, as can be seen in the general integration scheme in Figure 9. The use of Node-RED and MQTT protocols to define component behaviours and communications also facilitated the integration of components.

Figure 9 contains two blocks which can be highlighted and on which most of the system is mounted as follows:Raspberry Pi: A model 4 with 8 Gb of RAM memory to which a 240 Gb SSD hard drive has been added containing Home Assistant. On top of it, a generic Zigbee cc2531 and Conbee II gateway, a Mosquitto MQTT broker, a Node-RED server v.3.0.2 and a Home Assistant software package called Zigbee2MQTT has been used. Thus, it is possible to obtain information from the home automation sensors and the Empatica E4 device through MQTT.Embedded PC with ROS Master: An Intel NUC with Ubuntu 16.04 installed to run on ROS Kinetic. The logic and software that provide interoperability and transparency between the different elements of the system are executed on this PC. The websocket from the rosbridge_server package is executed for integration with Home Assistant to publish and subscribe to topics from Node-RED. Integration with other elements in the system implemented in ROS Kinetic is direct.

A specific “Himtae” package defined in ROS was used for communications between the elements of the system, with the messages to be exchanged according to the communication protocols between the different systems and the robots. The inclusion of the Echo 8 smart speaker allowed Alexa to be integrated with the rest of the system as a user–robot interface. The Raspberry Pi server was configured to receive Alexa requests to integrate any service with the backend of the Alexa skill developed: databases, home automation, third-party APIs and ROS. This flexibility gives the user information about room temperature, sending complex orders to the assistance robot through a user-friendly interface.

The domestic manipulator robot has a total of four modules that allow it to function correctly. The first, the base, allows the robot to navigate its environment. The second, the camera, is responsible for detecting objects found in the scene and identifying objects of interest. The third, the arm, is responsible for approaching an identified object for subsequent manipulation. The gripper module is responsible for gripping the object. All of these modules work individually, so it is necessary to program a controller that coordinates each module in use. A scheme of the proposed controller is shown in Figure 10.

This controller was designed in ROS as a state machine to toggle between the four modules, for which messages are proposed. For the base, it sends the desired pose to the robot. Once this pose is reached, the base module returns a Boolean to the controller, which starts up the next desired module. The controller specifies the object to be searched for by the camera and receives the spatial coordinates of the object in question. For the arms, the controller sends the desired end point and orientation of the end effector and receives a Boolean when it has been reached. The gripper then receives a Boolean from the controller indicating whether it should open or close, and after performing the corresponding action it sends a confirmation Boolean.

## 3. Results

In this section, several results obtained in experiments different to those performed in [44] are presented and discussed.

### 3.1. Data-Capture Experiment for Mood Prediction

Continuing with the experimentation dynamics explained in [44], with datasets of less than one month per subject, an attempt was made to capture a dataset of at least one month with several subjects. In this case, data were collected from two control participants with different digital capabilities to compare their response times. The first participant was a 63-year-old man in constant contact with technology. The second was a 75-year-old technologically inexperienced woman. The ultimate goal of these experiments was to obtain a large dataset (minimum 1 month) for each participant. However, due to the difference between their profiles, adjustments had to be made when defining the stages of these experiments. Since the first subject was an advanced user of technology, the system was simply explained to him, and he was asked to use the Empatica E4 device for at least 1 month and answer the questions on the Android application five times. As the second subject was a person who had little contact with technology, we devised a two-phase experiment that prioritized adaptability and learning to use the system:First Phase: In this phase, the user only had to answer the questions proposed by the Android application and carry the Empatica E4 device, without being required to connect or charge it. In this way, the user would gradually become accustomed to the acquisition system.Second Phase: In this phase, the user, in addition to answering the questionnaires proposed by the application, was asked to have the Empatica E4 device connected in order to start the data capture.

In the case of the first subject, the experiment lasted 47 days, while for the second subject it lasted a total of 114 days. On the first phase, a total of 65 days were spent, and on the second phase, a total of 49 days.

The data for the two subjects are shown below in the order given above and the results are discussed. As previously explained, the questionnaires contain two questions on an individual’s current level of activity or contentment. These levels are quantified in discrete sectors ranging from 0–4. The level of −1 is assigned for days on which this question has not been answered.

As can be seen in Figure 11 and Figure 12, the rate of use of the application was not very continuous. Upon analyzing the data, we realized that the subject did not answer the questionnaires on at least 44.6% of the days. As regards using the Empatica E4 medical device, we noticed that the data were similar. In Figure 13, we can see that we have records for 24 days, which is still around 50%. On the other hand, Figure 13 shows a good data-capture trend as regards the duration of the daily sessions. In general terms, after the first seven sessions, the time averages for the Empatica E4 device are 8 h or more, which is a good sign.

As can be seen, during the first phase (Figure 14 and Figure 15) of the study, there were many days when the user did not answer the proposed questionnaires.

As can be seen in the graph for the second phase (Figure 16), the experiment had a much stronger continuity. While there were 36 days with no response in the first phase (just over 50%), in the second phase, there was no response on only 4 days (8.16%).

### 3.2. Social Path-Planning Results

In this work, we tested the modified FM^2^ algorithm in order to obtain smooth paths that do not interfere with people in the home environment. When humans are present in the environment as individuals, not taking part in social interactions, the robot treats each one of them as a separate entity. Thanks to the design of FM^2^ and the personal-space model used, the robot will create safe, smooth and human-friendly trajectories to reach a goal, as shown in Figure 17.

On the other hand, when the robot needs to accompany the user, the human can be treated as the leader of the formation and the robot follows him/her according to the robot formation motion-planning algorithm. The algorithm tries to maintain this situation as far as possible. The way the algorithm is designed will give preference always to the human in the cases of narrow corridors or cluttered environments. A sequence resulting from the application of this algorithm is shown in Figure 18.

### 3.3. Path-Planning Manipulation Results

Experimental results for the path planning were generated using Matlab. FM^2^ and modified FM^2^ were applied to the end effectors of the arms, considering that the position of the wrist must be horizontal at the end of the trajectory in order to grasp objects. As explained, objects and users are avoided with FM^2^ and joint collisions are avoided using differential evolution. The results for a generated path (red) and final arm positions (green) are shown in Figure 19.

### 3.4. Results of a Global Test in Real Scenarios

Several test scenarios were implemented, for example, the “start of the day” scenario (Figure 20): when the alarm sounds in the morning, the lights progressively turn on and Alexa provides information about the day’s agenda, weather, etc. The assistant robot approaches the bed to suggest getting up to the user and asks if they want the robot in the kitchen to prepare the coffee. When the user is detected getting out of bed, the robot sends a message to the robot in the kitchen and either goes to a corner in safety mode to await instructions or goes to the kitchen to wait for instructions when the coffee is ready to pick up.

Once the manipulative robot has received the order to heat the coffee, it must plan the route and the gripping activity. First of all, the destinations are defined semantically. To obtain the geometric destinations, we use a semantic planner developed in [58] that establishes the geometric positions of the objects in a room. After modelling the environment, the starting points and the target (work zone) are established (points A and B in Figure 21).

The hybrid metric–topological planner is then launched on the map layers. An initial test was carried out in a simple scenario (see Figure 21). From its initial resting position, the robot was commanded to navigate inside a room until it approached a table, where the approaching point for the robotic base was an established point in Euclidean space. As no doors needed to be traversed, only the geometric part of the planner was activated. The resulting path proved the smoothness of the applied geometric planner, FM^2^.

When dealing with a more complex scenario in which doors need to be passed through, the topological side of the planner is applied, as shown in Figure 22. Initially, the geometric map is segmented, and the topological map is extracted. In another test, the robot was commanded to reach the same table as in the above test and was placed on topological node number 5, while the table was on node number 1. The final topological path and the local geometric paths were proven to speed up execution with respect to the traditional geometric global planners, achieving a robust navigation system for the manipulating robot.

The robot then activates the handling mode, uses the 3D scanner to reconnoitre the work area and launches the manipulator to pick up the cup of coffee and put it in the microwave by pressing the power button. Once heated, the robot opens the door, takes the cup of coffee and gives it to the personal assistance robot to take it to the user. Figure 23 shows the robot putting the cup of coffee into the microwave.

The assistance robot proposes activities, such as calling a relative, exercising, etc., during the day, according to the user’s detected emotional state. It also suggests going for a walk if it thinks the user is spending too much time in front of the TV, etc. It may also get the manipulative robot to reach for some utensil, medicine, etc., and put it in the assistance robot’s tray to take to the user.

## 4. Discussion

The overall aim of the HIMTAE system was to create a heterogeneous system incorporating two robotic platforms and Ambient Assisted Living to help elderly people who live alone at home in their daily activities and suggest activities to keep them positive and active. In this regard, the environment must be suited to the person, adapted to the user’s capabilities so as to help them maintain autonomy for longer with a higher quality of life. Both operating platforms act in a coordinated manner: the domestic-assistance robot was conceived as a two-arm manipulating robot to carry out tasks in the home, which requires the ability to move around a room as well as manipulation skills, taking account of obstacles and users using semantic concepts and social path planning. The mobile accompanying robot was developed to work together with the Ambient Assisted Living (AAL) system integrated in the home, the artificial intelligence algorithms and the biometric sensors incorporated in the medical bracelet.

The benefits of having a modular and scalable Multirobot and Ambient Assisted Living system were experienced in the different tests accomplished during the system’s development and testing. Thanks to the open architecture, different system deployments to help the elderly were tested, from the simplest model (with only home automation and smart speakers to communicate and interact with the assisted environment, possibly including the social robot) to the most complex model, which includes two robots, a social assistant and manipulator.

Due to the maturity of IOT technology and the decision of manufacturers to unify wireless protocols for home automation environments (Zigbee, Patter, etc.), it was easier to add new sensors to the system, which expands the field of information obtained from the user. All these sensors are integrated in a server inside the home, with all the necessary security protocols to prevent computer attacks. This server is based on an open-source operating system developed for home automation called Home Assistant (HA). This OS can run on almost any hardware platform, which makes the central node system “immune” to changes in microcomputer technology, such as Raspberry systems, Intel NUCs, etc. It is designed as a free platform for the integration of different commercial products and has a secure server in the cloud (Nabucasa). Running on HA, we have, among others, open communication standards, such as Mosquito Browser, databases, such as MariaDB, open representation tools, such as Grafana, and graphical programming frameworks, such as NodeRed, etc. The robots can also communicate thanks to the central node, where roscore runs, and subscribe to the necessary communication topics. The coordination of the whole system is implemented by NodeRed, taking into account the decisions of the AI algorithms.

There are an increasing number of works in the literature that propose the use of one or more robots for home assistance, as commented on in the review of the state of the art in Section 1. Many systems, such as those presented in [4,5], that offer solutions based on multirobot systems show that even people with reduced mobility can benefit from approaches based on multiple assistance robots, although it is more difficult to find an integrative solution like ours, with a multiple robot system that includes AAL and manipulation capabilities to provide complete home assistance (mental and physical).

The semantic segmentation of the navigation and planning environment in the mobile manipulator for navigation and movement in the manipulation area facilitates the tasks of the later planners. Although the current tasks carried out by the robots are predefined, the system must be provided with a navigation system that combines semantic concepts with the action-based planning paradigm to provide the system with a higher level of autonomy. At a low level, the FM^2^ algorithms achieve smooth and safe travel routes for the base and the manipulator, while differential evolution was shown to be ideal for avoiding arm collisions during manipulation. It is planned to improve handling by replacing two-fingered grippers with robotic hands and introducing ambidextrous handling algorithms.

For the development of the robotic platforms, ROS and proprietary algorithms such as those described in this article were used for efficient mapping and autonomous navigation in a real home environment. The home automation system was tested for a longer period of time and integrated in a real test environment. The number of subjects for estimating moods was expanded, as was the available dataset, to make it increasingly adaptable and reliable. These tests and results can be found in [44], which concluded that the model showed a promising ability to predict mood values using only physical data and machine-learning techniques.

Although the results presented in the previous section are satisfactory as regards navigation, manipulation and correct performance of the expected tasks, as well as predicting moods and suggesting activities, work still has to be done to enhance the system’s robustness and fault-tolerance, by improving, for instance, the robot autodocking algorithms and the ability to locate people and dynamic obstacles inside the house. Currently, we are performing new tests to improve the system while expanding the range of tasks that it can carry out. Artificial intelligence algorithms other than those used in [34,36] are being used to improve mood prediction. The method used by the HIMTAE system to gather information on mental well-being has been used in other works, such as [38], to monitor potential depressive patterns in elderly people living alone, and in [39], for a system that gathers information on a person’s activities and detects long-term stress patterns. We can thus conclude that the techniques are well used, but the long-term psychological impact of the suggested activities needs further study to verify the user’s overall improved mental health.

It should be noted that the current mood-prediction system is equipped only with physiological data. This is why we have applied for funding to continue the development in order to combine different data modalities, such as information on behaviour inside the house deduced from home automation data, use of mobile phones, identification of gestures and voices, etc. It is also necessary to provide better monitoring of the subject’s mood and mental state, together with the implementation of interventions (guidance and psychological counselling) to improve mental well-being. Progress will have to be made in the healthcare system in response to the demands of today’s society, in which the promotion of digital health has become a priority (Mental Health Strategy of the National Health System 2022–2026). It is therefore necessary to adapt services to the needs of the population, giving each person greater autonomy and capacity, as well as ensuring the sustainability of systems.

To sum up, it can be concluded that this is an innovative proposal aimed at empowering the elderly to be more independent in their own homes and improve their quality of life.

## Figures and Tables

**Figure 1 sensors-22-07983-f001:**
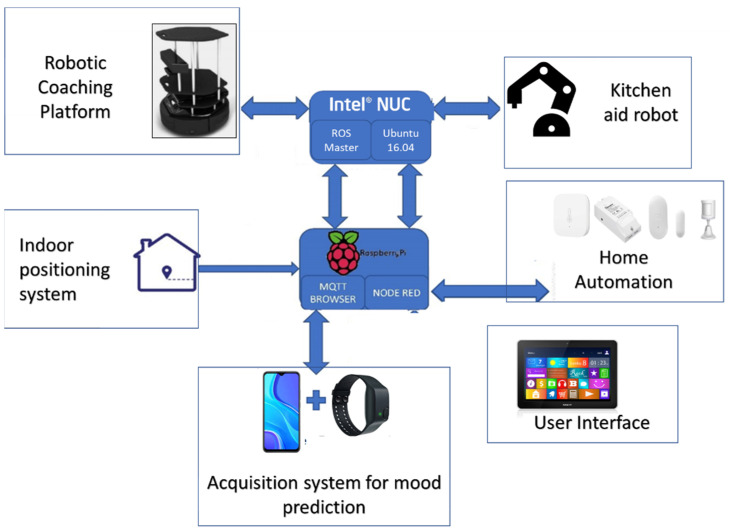
General architecture of the HIMTAE heterogeneous system.

**Figure 2 sensors-22-07983-f002:**
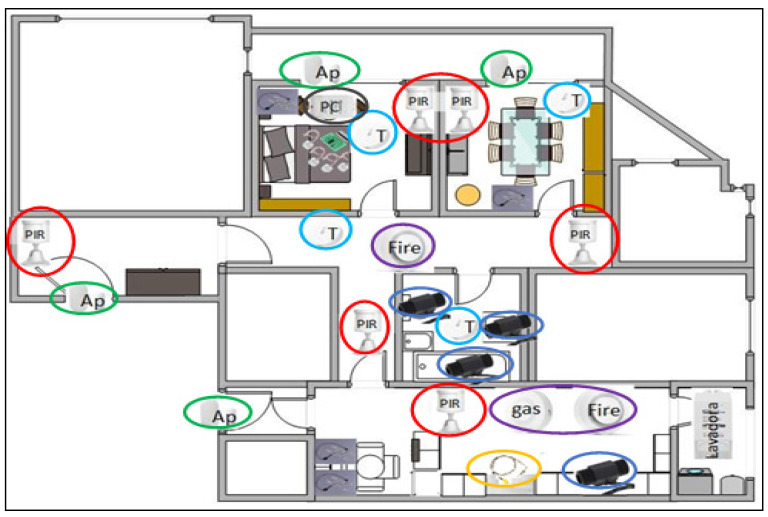
Distribution of home automation sensors. (AP = Opening/green, PIR = Presence/red, T = Temperature/light blue, PC = Electric consumption/black, gas = Natural gas/violet).

**Figure 3 sensors-22-07983-f003:**
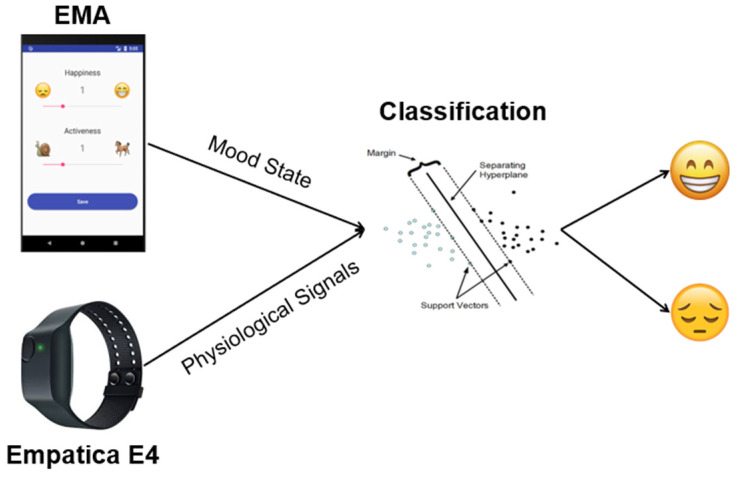
Machine-learning classification method of training the mood estimation algorithm [15].

**Figure 4 sensors-22-07983-f004:**
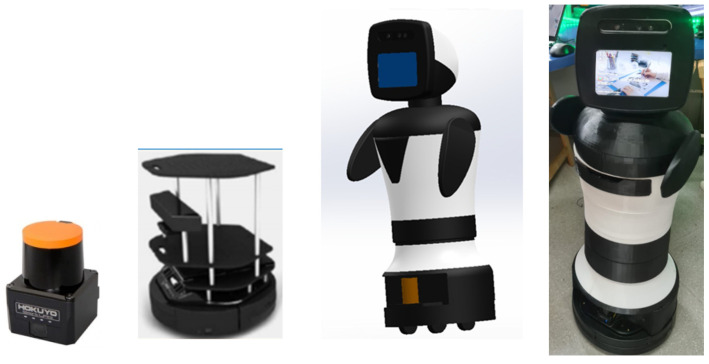
Assistive mobile robot: Turtlebot base and Hokuyo LIDAR (**left**), 3D model (**middle**) and real prototype (**right**).

**Figure 5 sensors-22-07983-f005:**
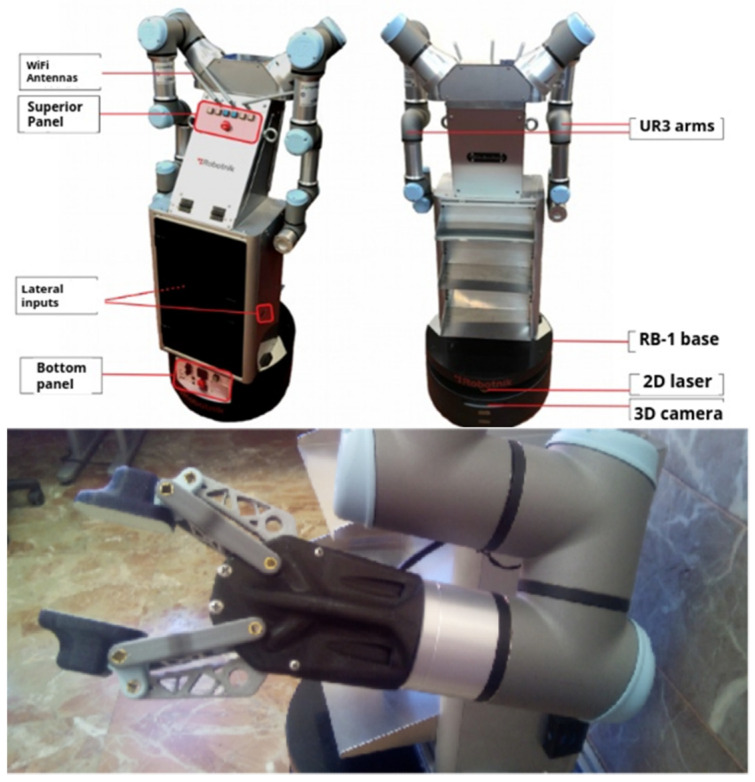
ADAM twin-arm manipulator robot and gripper.

**Figure 6 sensors-22-07983-f006:**
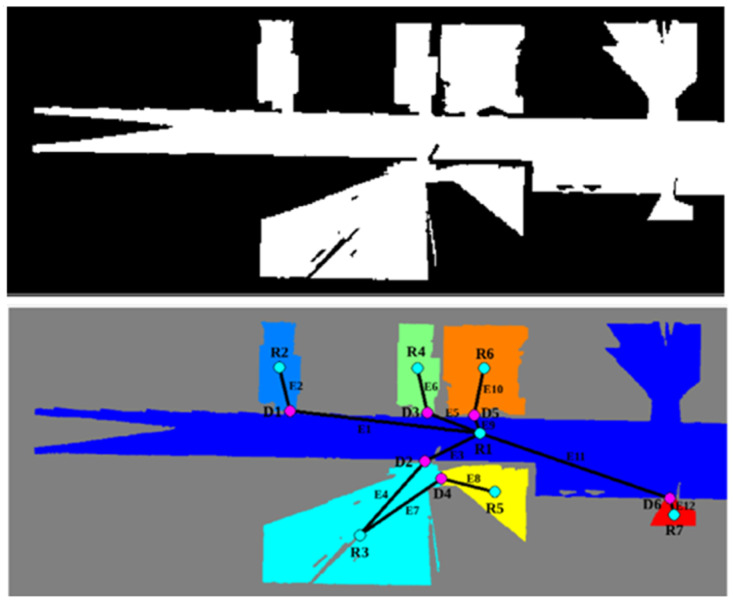
Segmented occupancy grid map and topological map extraction. Each node and edge is labeled with a different number, where R corresponds to room nodes, D corresponds to door nodes and E represent edges.

**Figure 7 sensors-22-07983-f007:**
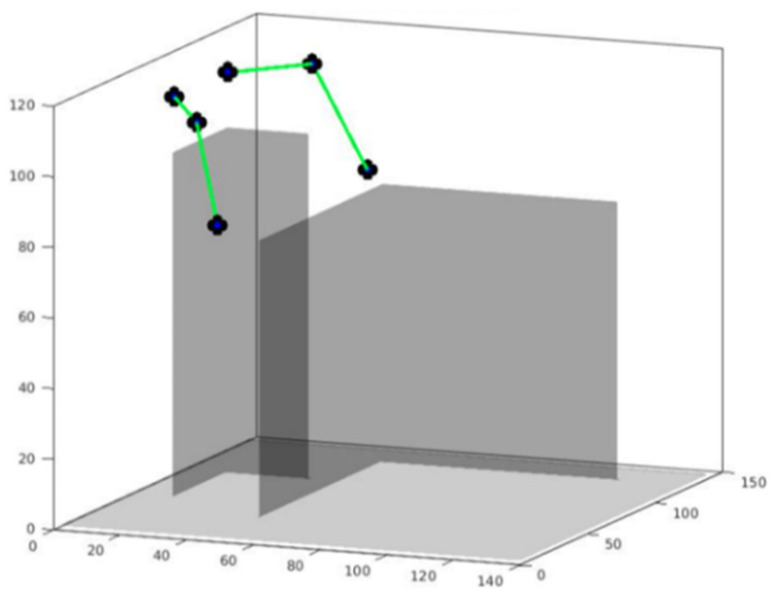
Simulated environment of the manipulative robot (**left**) and a table (**right**).

**Figure 8 sensors-22-07983-f008:**
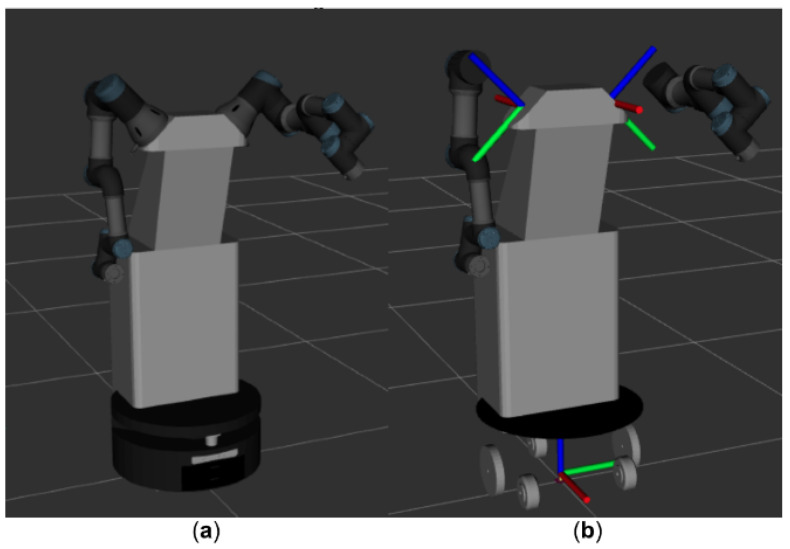
Robot visualized in Rviz: (**a**) complete robot model; (**b**) reference axes of the robot base and of its two arms.

**Figure 9 sensors-22-07983-f009:**
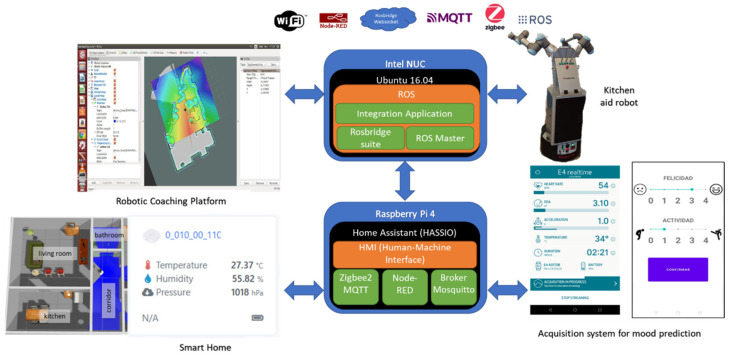
General integration scheme.

**Figure 10 sensors-22-07983-f010:**
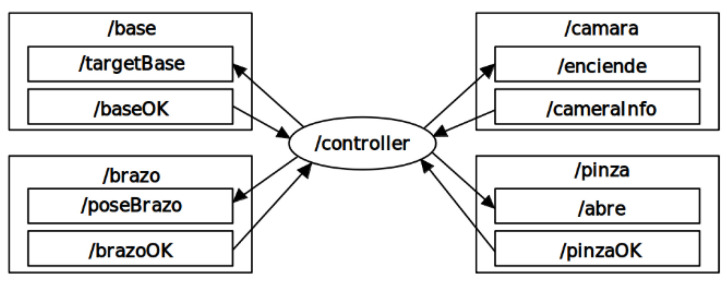
Structure of the controller and the four modules that define the operation of the physical assistance robot.

**Figure 11 sensors-22-07983-f011:**
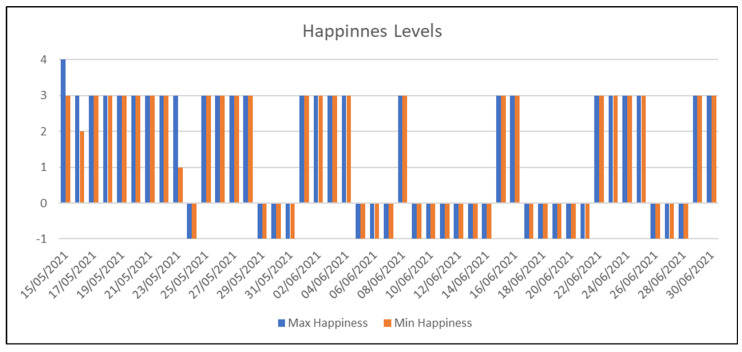
This figure shows the maximum and minimum happiness values recorded each day for the first subject.

**Figure 12 sensors-22-07983-f012:**
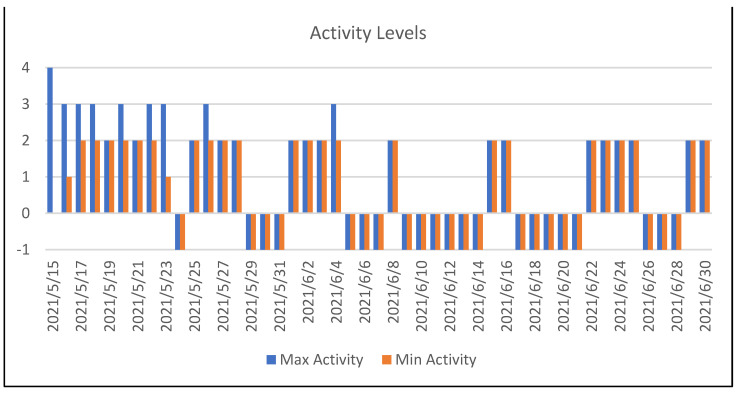
This figure shows the maximum and minimum activity values recorded each day for the first subject.

**Figure 13 sensors-22-07983-f013:**
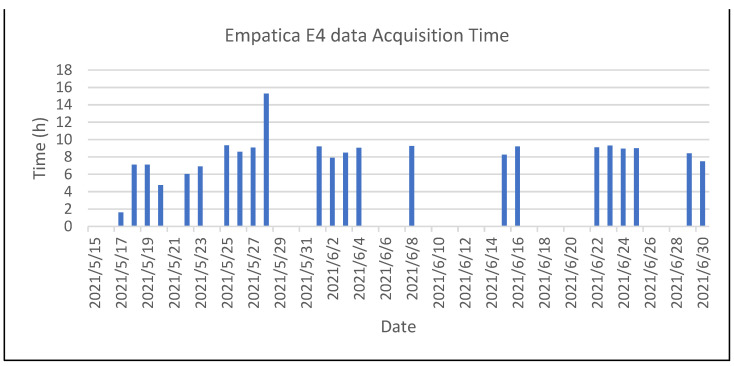
Duration of sessions recorded with the Empatica E4 medical device for the first subject.

**Figure 14 sensors-22-07983-f014:**
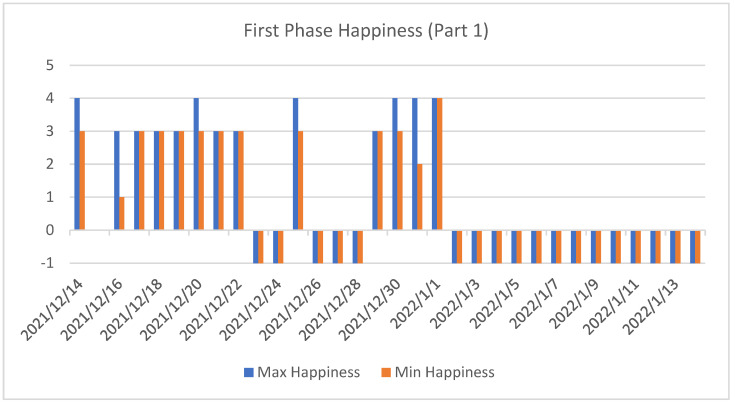
First set of data from the first phase of the experiment. This figure shows the maximum and minimum happiness values recorded each day for the second subject.

**Figure 15 sensors-22-07983-f015:**
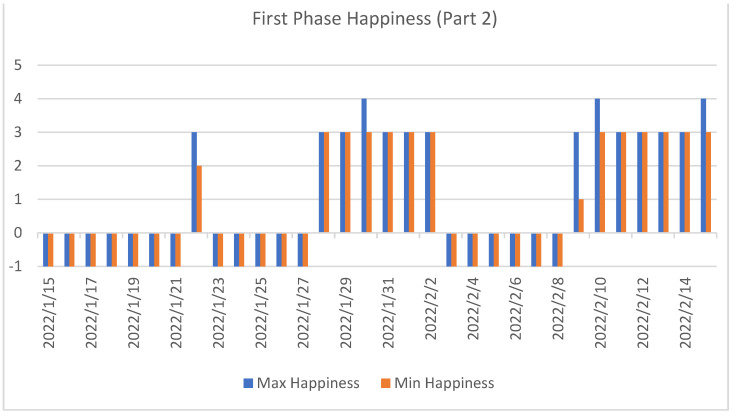
Second set of data from the first phase of the experiment. This figure shows the maximum and minimum happiness values recorded each day for the second subject.

**Figure 16 sensors-22-07983-f016:**
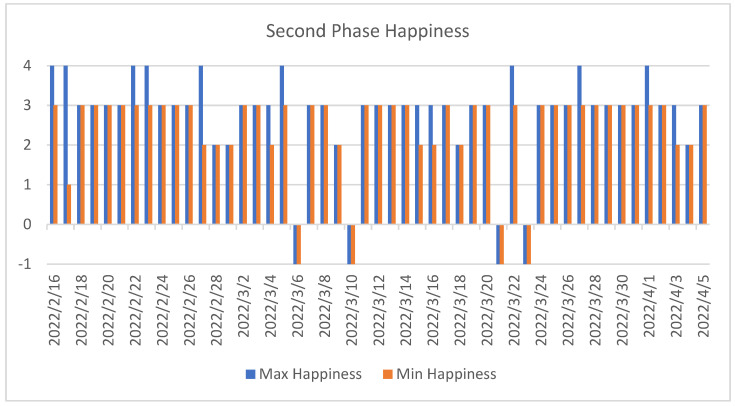
Second phase of the experiment. This figure shows the maximum and minimum happiness values recorded each day for the second subject.

**Figure 17 sensors-22-07983-f017:**
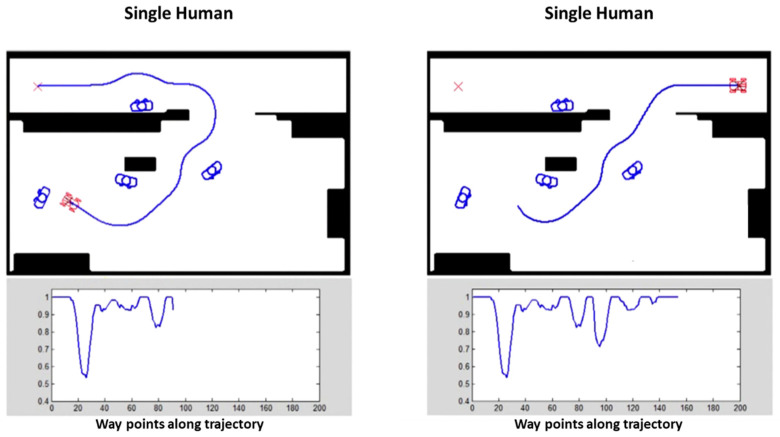
Results for the path-planning approach. The start point is at the top of the map and the path selected avoids personal spaces, except when it means to get very close to obstacles. The *y*-axis represents the velocity of the robot and the *x*-axis represent the trajectory of the robot.

**Figure 18 sensors-22-07983-f018:**
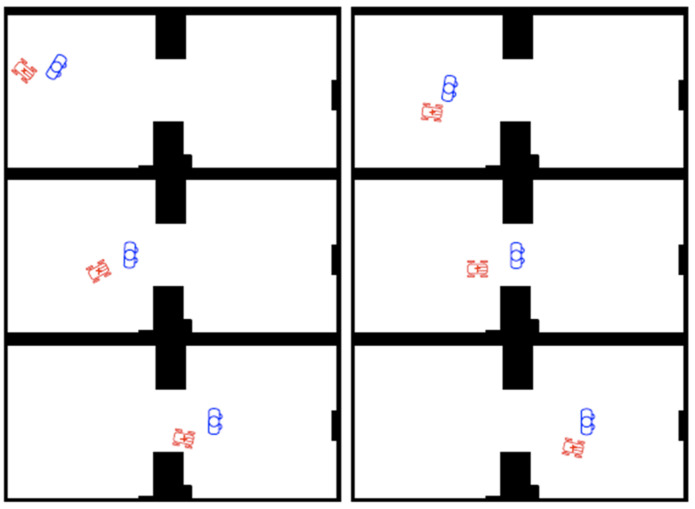
From top-left to bottom-right: sequence of a robot following a human with the FM^2^-based robot formation motion-planning algorithm.

**Figure 19 sensors-22-07983-f019:**
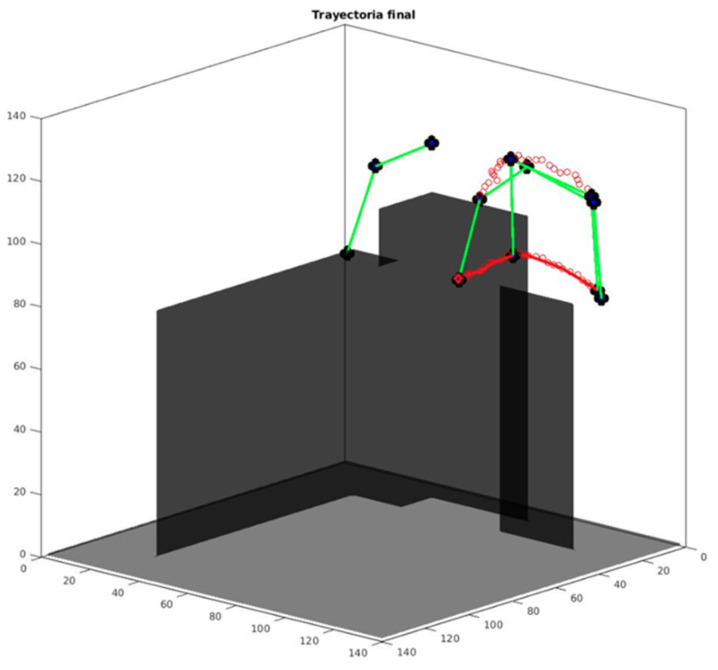
A generated trajectory. Red points are the paths calculated for the joints of the simplified arm.

**Figure 20 sensors-22-07983-f020:**
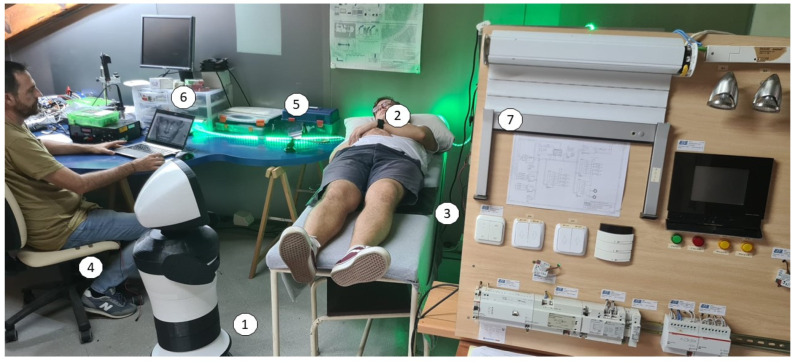
“Start of the day” test scenario: (1) Echo robot: 3D-printed Turtlebot base with Echo 8 and Alexa on head; (2) subject wearing the Empatica E4; (3) presence detector in bed; (4) presence detector in chair; (5) AAL user interface; (6) simulation of attention-status detection when viewing TV; (7) simulator of other home automation elements.

**Figure 21 sensors-22-07983-f021:**
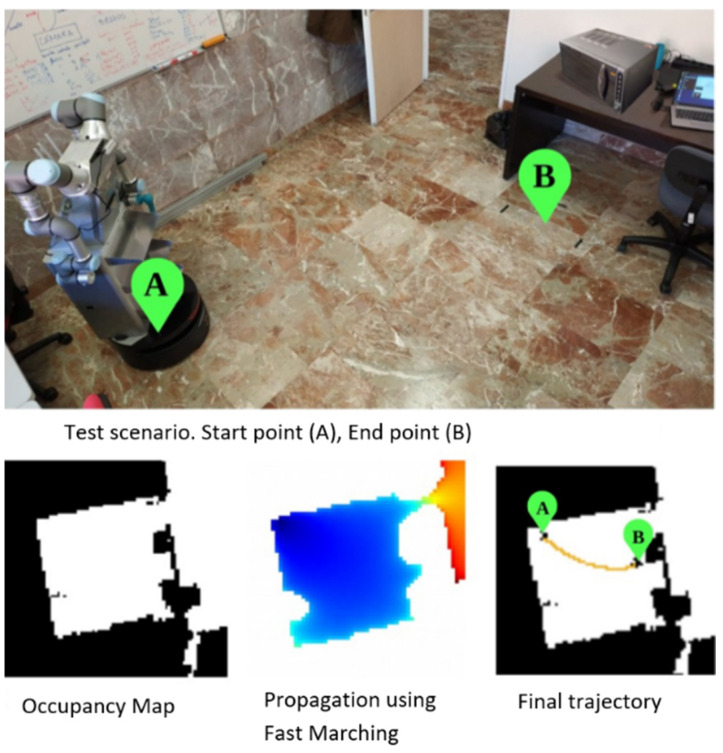
Example of planning with FM^2^, where the robot is at point A and must move to point B.

**Figure 22 sensors-22-07983-f022:**
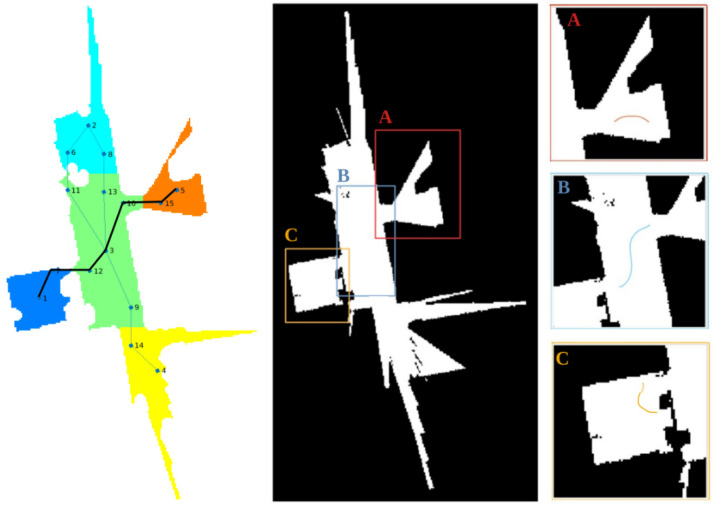
Complex scenario in which the robot needs to navigate through multiple rooms. The topological path is first computed (**left**, where nodes are numbered in order of appearance) and the geometric path planner is applied locally to each of the rooms (**right**, where rooms are labeled from start to end locations).

**Figure 23 sensors-22-07983-f023:**
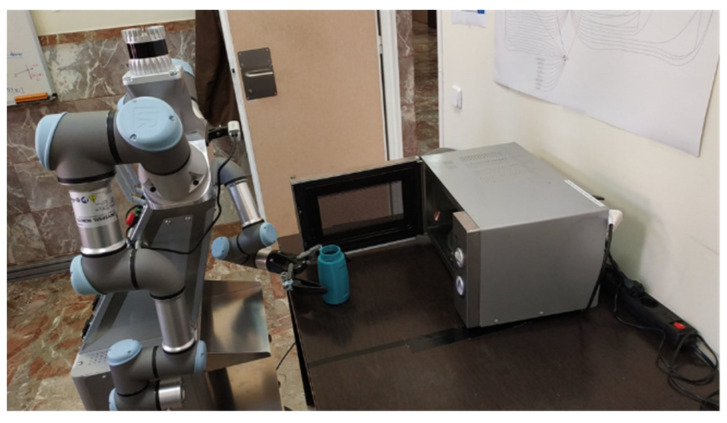
Manipulator robot inserting the cup of coffee into the microwave.

**Table 1 sensors-22-07983-t001:** Information collected by the Android.

Information	Description
Happiness level	Representation of the happiness level felt by the user at the time of the test on a discrete scale divided into 5 levels (0–4).
Activity level	Representation of the activity level felt by the user at the time of the test on a discrete scale divided into 5 levels (0–4).
Test hours	User-defined time register of time tests.
Reaction time	Timestamp at which the application gives notification that the test is to be performed and the timestamp at which the user accesses the notification.

## Data Availability

The data presented in this study are available on request from the corresponding author. The data are not publicly available due to privacy restrictions.

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
