# Peer review of "A Multirobot System in an Assisted Home Environment to Support the Elderly in Their Daily Lives"

_sensors, 2022, doi:10.3390/s22207983_

Round 1
Reviewer 1 Report
The paper deals with interesting and challenging problems in assistive robotics. It proposes nice ideas and combines different techniques. In general, I deem it well written, but some parts could be improved.
My main concerns are:
1. I do not find the motivation for using two robots strong enough. The multi-robot idea is highlighted as a significant contribution, but the advantages over using the robot with manipulation capabilities + one or several voice assistants or even static personal robots are not clear to me.
People with reduced mobility can also benefit from approaches based on multiple assistance robots -> Why?
2. Experiments and results are simple and limited. The manipulation example is really inspiring and could be super helpful, but the mobile robot trajectory is short, with no obstacles around, and the scenario has a simple and prepared setup. Results from this experiment are purely qualitative.
You explain the robot navigation strategy to go through doors but this is not shown at all in the experiments.
There are more advanced assistive robots which have been tested in more realistic conditions. Some citations are missing in this regard. And I wonder if there is a specific reason for not mentioning PDDL.
Along the paper, some paragraphs seem a bit ad-hoc, focusing on details about the HIMTAE project (e.g. project partners) instead of on general contributions.
Other details:
- aims to be as intrusive as possible as much as possible.
- Resolution of Fig. 10 should be improved
- How is the navigation goal defined for manipulation tasks? (Ln 523).
- Experiment 4.1 with only one subjects seems limited.
- The graphs in Fig. 16 do not have labels and they are not mentioned in the caption or the text.
-The navigation example in Fig 19 is too simple. More advanced results have been shown in previous works.
- ref 44 and 18 seem to be the same
- Reference [19] is very nice but there are also more recent methods
- Evolution of a Cognitive Architecture for Social Robots: Integrating Behaviors and Symbolic Knowledge
- Impact of decision-making system in social navigation
Author Response
Please see the attachement. Following the responses please find a new version of the article with changes in red colour.

Reviewer 2 Report
The manuscript is interesting but must be improved.
I have a few comments for the authors:
1. abstract: better summarize the sections and avoid the use of “we”
2. Insert the methods section describing the design of the article
3. Insert a flow chart in the methods allowing the reader to better understand the design of the acrticle
4. there is a preliminary part of the study dedicated to a review (section 2 and a part of section 3). Insert in the design in the methods
5. Discussion must be alone
4. Introduce in the discussion the limitations and expand the comparisons to other studies
5. Text must follow the MDPI guidelines
6. Figures must be improved with labels and in resolution
Author Response

(The authors gave the same response as above.)

Round 2
Reviewer 1 Report
Thank you very much for your work and for your clarifying responses. I think that the article has been significantly improved.
The new text introduced some English mistakes, please try to carefully review the language.
I still think that a experiment with only 2 subjects is very limited, but I value the effort and its significance. I hope it is possible to extend this work in the future.
Reviewer 2 Report
N/A